# Influence of Alcohol Consumption on the Development of Erosive Esophagitis in Both Sexes: A Longitudinal Study

**DOI:** 10.3390/nu14224760

**Published:** 2022-11-10

**Authors:** Masahiro Sogabe, Toshiya Okahisa, Miwako Kagawa, Hiroyuki Ueda, Kaizo Kagemoto, Hironori Tanaka, Yoshifumi Kida, Tetsu Tomonari, Tatsuya Taniguchi, Hiroshi Miyamoto, Yasushi Sato, Masahiko Nakasono, Tetsuji Takayama

**Affiliations:** 1Department of Gastroenterology and Oncology, Tokushima University Graduate School of Biomedical Sciences, Tokushima 770-8503, Japan; 2Health Service, Counseling and Accessibility Center, Tokushima University, Tokushima 770-8502, Japan; 3Department of Internal Medicine, Shikoku Central Hospital of the Mutual Aid Association of Public School Teachers, Shikokuchuo 779-0193, Japan; 4Department of Internal Medicine, Tsurugi Municipal Handa Hospital, Tsurugi 779-4401, Japan

**Keywords:** erosive esophagitis, alcohol intake, development, sex, age

## Abstract

The influence of changes in alcohol consumption on erosive esophagitis (EE) development in both sexes is unclear. This observational study investigated sex differences in the influence of alcohol consumption on EE development, and included 2582 patients without EE at baseline from 13,448 patients who underwent >2 health check-ups over >1 year. The rates of non-drinkers who started drinking, and drinkers who abstained from drinking, who increased, and who decreased their weekly alcohol consumption were 7.2%, 9.7%, 14.7%, and 24.1% and 7.3%, 17.8%, 12.8%, and 39.0% in men and women, respectively. In the final cohort, 211/1405 (15.0%) men and 79/1177 (6.7%) women newly developed EE. The odds ratio (OR) for drinking in EE development was 1.252 (95% confidence interval (CI), 0.907–1.726) among men and 1.078 (95% CI, 0.666–1.747) among women. Among men aged <50 years, the OR for drinking ≥70 g/week in EE development was 2.825 (95% CI, 1.427–5.592), whereas among women, the OR for drinking ≥140 g/week in EE development was 3.248 (95% CI, 1.646–6.410). Among participants aged <50 years, the OR for daily drinking in EE development was 2.692 (95% CI, 1.298–5.586) among men and 4.030 (95% CI, 1.404–11.57) among women. The influence of alcohol consumption on EE development differed between the sexes. We recommend no alcohol consumption for individuals aged <50 years to avoid EE development. Daily drinkers should be assessed for EE development.

## 1. Introduction

Gastroesophageal reflux disease (GERD), classified into erosive esophagitis (EE) and nonerosive reflux disease, is one of the most common upper gastrointestinal disorders worldwide and may decrease quality of life [1]. Recently, the prevalence of GERD, including EE, has been increasing [2,3]. Similarly, the incidence of GERD has been increasing annually, although incidence estimates differ across regions: 18.1–27.8% in North America, 8.8–25.9% in Europe, and 8.8–7.8% in East Asia [4]. EE is a major risk factor for Barrett’s esophagus (BE) [5], which is the precursor to esophageal adenocarcinoma (EAC). Therefore, prevention of EE is important for the prevention of EAC. Although the influence of alcohol intake on EE development is known to differ between the sexes [6,7,8], and harmful alcohol use is recognized globally as the leading risk factor for morbidity, disability, and mortality [9], the role of alcohol in EE development is controversial and the influence of changes in alcohol consumption on EE development in both sexes remains unclear. Therefore, we investigated the association between changes in alcohol consumption and the presence of EE in both sexes. We used a longitudinal design to clarify the influence of alcohol consumption, including quantity and frequency, on EE development according to sex. We hypothesized that the risk of EE development would increase with an increase in alcohol consumption in older patients to a greater degree than that in younger patients.

## 2. Materials and Methods

### 2.1. Design and Study Population

In this observational, longitudinal study, we enrolled 13,448 patients who underwent regular, comprehensive health check-ups, including laboratory examinations and esophagogastroduodenoscopy, at Shikoku Central Hospital of the Mutual Aid Association of Public School Teachers (Shikokuchuo, Japan) from April 2015 to March 2020 (see Appendix A, a flow diagram of individuals undergoing health check-ups). Patients who had undergone regular health check-ups >2 times over an interval of >1 year during the study period were included. At baseline and in the final cohort, patients were excluded if they had incomplete information, had undergone previous digestive tract surgery, had visited the hospital for treatment, were followed up for upper intestinal disease (e.g., GERD, peptic ulcer, or upper intestinal cancer), were taking medications such as H2-receptor antagonists or proton pump inhibitors, or were diagnosed with upper gastrointestinal cancer at the time of esophagogastroduodenoscopy. Additionally, patients diagnosed with EE at baseline were excluded. The initial and most recent data from the same patients in the cohort were used as the baseline and follow-up data, respectively. Finally, 2582 patients were eligible. The study protocol conformed to the ethical guidelines of the 1975 Declaration of Helsinki. The Shikoku Central Hospital of the Mutual Aid Association of Public School Teachers’ institutional review board committee approved the study protocol (approval no.: H27-3, approval date: 1 March 2015). An opt-out approach was used to obtain informed consent from patients, and personal information was protected during data collection.

### 2.2. Data Collection

The following information was obtained using a self-report questionnaire: smoking status, alcohol intake, exercise, meals, drug history (including acid-inhibiting medication), and history of the present illness. Current smokers excluded individuals with a previous smoking habit. The amount of alcohol (in grams) consumed per drinking day was determined using the representative percent alcohol by volume for each type of alcohol: 5% for beer, 16% for Japanese sake, 25% for shochu, 10% for wine, and 34% for whisky. The average weekly alcohol intake was classified into five categories: none, 0.1–69.9, 70–139.9, 140–279.9, and ≥280 g/week. Drinking frequency was classified into three categories: non-drinking, occasional drinking (1–6 days/week), and daily drinking. Regular exercise was defined as performing a >30-min exercise session at least once weekly. The habit of “eating before going to bed” was defined as eating within 2 h of going to bed at least weekly. Anthropometric parameters such as height, weight, and waist circumference (WC) were recorded for all patients. A body mass index (BMI) cut-off of ≥23 kg/m^2^ was used to define overweight and obesity because all patients were Japanese [10]. Blood samples were obtained from all patients in the morning after 12 h overnight fasting. Clinical laboratory tests were performed to measure the serum levels of aspartate aminotransferase, alanine aminotransferase, gamma-glutamyl transpeptidase, total cholesterol, high-density lipoprotein cholesterol (HDL-C), triglycerides (TGs), low-density lipoprotein cholesterol, uric acid, fasting plasma glucose (FPG), hemoglobin A1c, and high-sensitivity C-reactive protein. Hypertension was defined as a blood pressure of ≥130/85 mmHg or the use of medications for hypertension. Dyslipidemia was defined as a TG level of ≥150 mg/dL or an HDL-C level of <40 mg/dL for men and <50 mg/dL for women or the use of medications for dyslipidemia. Impaired glucose tolerance (IGT) was defined as an FPG level of ≥100 mg/dL or the use of medications for diabetes mellitus (DM). Evaluation of *Helicobacter pylori* infection is described in Appendix A.

### 2.3. Esophagogastroduodenoscopy

A standard endoscopic examination of the esophagus, stomach, and duodenum was performed by endoscopy specialists with >5 years of experience in endoscopy. Esophagogastroduodenoscopy was performed using a conventional single-channel endoscope (GIF-H290, -H260, -PQ260, -XP260, or -XP260N; Olympus, Tokyo, Japan). The severity of EE was graded from A to D according to the Los Angeles (LA) classification. Endoscopic findings from each patient were independently validated by a single endoscopy specialist. Newly developed EE was defined as a diagnosis of EE following a diagnosis of no EE at baseline.

### 2.4. Statistical Analysis

Continuous variables are expressed as means ± standard deviations. Categorical data are expressed as counts (percentages). Categorical variables were compared between two groups and more than two groups by using the χ^2^ test and m × n χ^2^ test, respectively. As the continuous data were not normally distributed, the Mann–Whitney U and Kruskal–Wallis nonparametric tests were performed to compare two groups and more than two groups, respectively. Differences were considered statistically significant at *p* < 0.05. A generalized estimating equation (GEE) can be used to manage the longitudinal data of patients who share common characteristics [11]. A GEE with a logit link and binomial distribution was used to describe the association between drinking status (i.e., average weekly drinking quantity and drinking frequency) and newly developed EE in the longitudinal analysis. In the first analysis, we investigated the influence of alcohol consumption on newly developed EE, adjusting for age, BMI, and WC. In the second analysis, we adjusted for lifestyle habits such as current smoking, regular exercise, eating before going to bed, and eating breakfast. In the third analysis, we adjusted for variables related to metabolic dysregulation such as hypertension, dyslipidemia, and IGT. In the fourth analysis, we adjusted for variables related to age, BMI, WC, lifestyle habits, metabolic dysregulation, and *H. pylori*. Odds ratios (ORs) and 95% confidence intervals (CIs) were calculated. All statistical analyses were performed using SPSS for Windows (version 27.0; IBM Corporation, Armonk, NY, USA).

## 3. Results

### 3.1. Baseline Characteristics According to Sex

The prevalence of smokers and drinkers; lifestyle habit-related factors; metabolic dysregulation-related factors such as hypertension, dyslipidemia, and IGT; and liver enzymes were significantly higher in men than in women (Table 1).

### 3.2. Changes in Drinking Quantity between Baseline and Final Measurements

Among men, 26 (7.2%) of 361 non-drinkers at baseline had started drinking (Table 2). Of 1044 drinkers at baseline, 101 (9.7%) men abstained from drinking, whereas 153 (14.7%) increased and 252 (24.1%) decreased their weekly drinking quantity across drinking categories. Among women, 52 (7.3%) of 717 non-drinkers at baseline had started drinking. Of 460 drinkers at baseline, 82 (17.8%) women abstained from drinking, whereas 59 (12.8%) increased and 142 (39.0%) decreased their weekly drinking quantity across drinking categories.

### 3.3. Changes in Drinking Frequency between Baseline and Final Measurements

Among men, 1149 (81.8%) had not changed their drinking frequency, whereas 89 (6.3%) increased and 167 (11.9%) decreased their drinking frequency across the drinking frequency categories (Table 3). Among women, 1003 (85.2%) had not changed their drinking frequency, whereas 74 (6.3%) increased and 100 (8.5%) decreased their drinking frequency across the drinking frequency categories.

### 3.4. Presence or Absence of EE in Patients without EE at Baseline

Among men, 211 (15.0%) of 1405 patients without EE at baseline developed EE at follow-up (Table 4). Among men with new EE, 86.3% had grade A EE. Among women, 79 (6.7%) of 1177 patients without EE at baseline developed EE at follow-up; the prevalence of grade A EE was 91.1%. The prevalence of EE development was not significantly different across male age groups, but was significantly different across female age groups (*p* < 0.05) (see Appendix A, illustrating the comparison across age groups of the prevalence of EE development according to sex).

### 3.5. Prevalence of EE According to Drinking Quantity and Frequency Categories

There were no significant differences in the prevalence of new EE onset according to drinking quantity and frequency categories among men; however, the prevalence of new EE onset was significantly different across drinking quantity categories among women (*p* < 0.005) (Figure 1). 

### 3.6. Relationship between EE Development and Alcohol Quantity

After adjusting for age, BMI, WC, lifestyle habits, hypertension, dyslipidemia, IGT, and *H. pylori*, the OR for EE development in male drinkers was 1.252 (95% CI, 0.907–1.726, *p* = 0.171) (Table 5). For men aged <50 years, drinking ≥70 g/week was a significant risk factor for EE development (OR = 2.825, 95% CI, 1.428–5.542, *p* < 0.005). Among women, after adjusting for age, BMI, WC, lifestyle habits, hypertension, dyslipidemia, IGT, and *H. pylori*, the OR for EE development in drinkers was 1.078 (95% CI, 0.666–1.747, *p* = 0.760). Drinking ≥140 g/week was a significant risk factor for EE development among women (OR = 3.248, 95% CI, 1.646–6.410, *p* < 0.005).

### 3.7. Relationship between EE Development and Drinking Frequency

Among men, after adjusting for age, BMI, WC, lifestyle habits, hypertension, dyslipidemia, IGT, and *H. pylori*, the ORs for EE development were 1.294 (95% CI, 0.896–1.867, *p* = 0.280) and 1.215 (95% CI, 0.853–1.731, *p* = 0.169) in occasional and daily drinkers, respectively (Table 6). Among women, after adjusting for age, BMI, WC, lifestyle habits, hypertension, dyslipidemia, IGT, and *H. pylori*, the ORs for EE development were 0.690 (95% CI, 0.374–1.273, *p* = 0.235) and 2.444 (95% CI, 1.320–4.524, *p* < 0.005) in occasional and daily drinkers, respectively. Among patients aged <50 years, the ORs for EE development were 2.692 (95% CI, 1.298–5.586, *p* < 0.01) and 4.030 (95% CI, 1.404–11.57, *p* < 0.01) in male and female daily drinkers, respectively.

### 3.8. Relationship between Alcohol Consumption and EE Severity in Patients with Newly Developed EE

No relationship between alcohol quantity and EE severity was noted in both men and women (see Appendix A, summarizing the relationship between alcohol quantity and EE severity according to sex). However, in the comparison of the prevalence of severe EE according to drinking quantity categories, there was a nonsignificant trend in men for the increasing prevalence of severe EE (defined as LA grades B, C, and D) as alcohol consumption increased (see Appendix A, illustrating the prevalence of severe EE according to drinking quantity categories and sex).

No relationship between drinking frequency and EE severity was noted in both men and women (see Appendix A, summarizing the relationship between drinking frequency and EE severity according to sex). In the comparison of the prevalence of severe EE according to drinking frequency categories, there was no relationship between the prevalence of severe EE (defined as LA grades B, C, and D) and drinking frequency for either men or women (see Appendix A, illustrating the prevalence of severe EE according to drinking frequency categories and sex).

## 4. Discussion

The principal results were that, among male patients aged <50 years, alcohol consumption of ≥70 g/week and daily drinking were risk factors for EE development, and, among female patients, alcohol consumption of ≥140 g/week and daily drinking were risk factors for EE development. Although the influence of alcohol consumption on EE development differed between men and women, drinking status by age is an important consideration for the prevention of EE development.

In several recent systematic reviews, an association between *H. pylori* infection and the development of GERD, including EE, has been reported [12]. Zamani et al. reported that *H. pylori* infection is associated with decreased odds of GERD symptoms and EE development [12]. In addition, *H. pylori* eradication was an environmental risk factor for EE in observational studies and randomized controlled trials [13]. The mechanism for an inverse association of *H. pylori* serostatus with EE may be the generation of ammonia, decreased acid production because of gastric atrophy, and a neuroimmunological influence [14]. In the present study, *H. pylori* infection was also significantly associated with a decrease in the prevalence of EE (see Appendix A, summarizing the ORs for EE development in each category of alcohol quantity and other factors).

Although the relationship between EE development and laboratory parameters associated with lifestyle-related diseases, such as DM, dyslipidemia, and hypertension, is controversial, a positive relationship between these parameters and EE have been more frequently reported than an inverse relationship [15,16,17,18]. Most patients with DM with esophageal dysfunction display evidence of coexistent peripheral motor or autonomic neuropathy. Although the pathophysiology of esophageal dysfunction that may induce GERD in patients with DM is unclear, DM-related esophageal dysfunction may be caused largely by autonomic neuropathy, especially vagal nerve damage [19]. In addition, delayed gastric emptying owing to DM-related autonomic neuropathy may promote the onset of EE [20]. Both esophageal and lower esophageal sphincter functions were unaffected when the serum concentration of alcohol was <70 mg/dL in a previous report [21]. In the present study, IGT was a significant risk factor for EE development in men (OR = 1.373, 95% CI, 1.010–1.882, *p* < 0.05) (Appendix A).

Although the influence of alcohol consumption on EE development has been suggested to vary according to alcohol type, many reports have concluded that drinking is a risk factor for EE development, regardless of alcohol type. For example, high liquor intake reportedly increases the EE risk up to two-fold [22]. An inverse relationship between alcohol consumption and EE development has also been reported. Anderson et al. reported that, although there was no association between total alcohol consumption and EE development, the consumption of wine was inversely associated with the onset of EE (OR = 0.45, 95% CI, 0.27–0.75) [22,23]. Kubo et al. reported that light beer drinking was inversely associated with the severity of GERD symptoms [24]. Additionally, a modest intake of red wine has been associated with a reduced risk of EAC [25,26]. Antioxidant agents in wine and beer may induce an inverse association between drinking and EE development [27,28], as the pathological changes associated with EE are caused by the activation of inflammatory pathways by reflux substances, which leads to mucosal damage [29]. Although several reports have indicated that lifetime alcohol intake is associated with BE [30], this relationship remains controversial [23,31]. The relationship between lifetime alcohol intake and EE development was not investigated in the present study. However, the prevalence of drinkers with newly developed EE was (169/211) 80.1% and (35/79) 44.3% in men and women, respectively. In addition, the prevalence of women with newly developed EE significantly increased as alcohol quantity increased, and there was a trend for the prevalence of men with new EE to increase with increasing alcohol quantity. Further investigations on the relationship between lifetime alcohol intake or the age at the onset of alcohol consumption and development of EE are needed.

In the present study, alcohol consumption and daily drinking were significant risk factors for EE development among men aged <50 years, but not among those aged ≥50 years. We had initially assumed that alcohol consumption may become a risk factor for EE development in older patients rather than in younger patients, based on declining alcohol metabolism and increasing total alcohol consumption with increasing age. In addition, age-related decreases in esophageal defense mechanisms, such as esophageal mucosal resistance to acid exposure and esophageal motor function clearing acid to the stomach, may be related to EE development [32]. The prevalence of GERD in adults reportedly increases with age [32,33]. In the present study, for men aged <50 years, diet, sleep status, work pressure, and an understanding of health management may influence EE development more than that for those aged ≥50 years. Alcohol consumption of ≥140 g/week and daily drinking were also risk factors for EE development among women in the present study. These distinct results may be related to the known differences in the influence of alcohol intake between the sexes [6,7]; alcohol-related liver disease is more common in men, and women are more easily affected by alcohol than men [34,35].

The strengths of the present study are the large cohort size, EE diagnosis by endoscopy specialists, longitudinal design, and assessment of newly developed EE according to drinking status, including quantity and frequency. Nevertheless, the study has several limitations. First, selection bias is possible because the participants were sufficiently conscious of their health to undertake a self-paid medical check-up; most participants were office workers of middle and high socioeconomic status. Therefore, the participants in the present study may not be representative of the general population. Additionally, whether a study of patients hospitalized for EE would yield similar results remains unclear. Second, the definition of a negative *H. pylori* infection status in the present study was strict. Therefore, false-positive results for *H. pylori* infection status were possible. Third, the genotype of aldehyde dehydrogenase 2, which is associated with alcohol metabolism and levels of estrogen-related sex hormones associated with menses, was not assessed as it is not generally conducted during medical check-ups. Fourth, we excluded patients taking acid-inhibiting medications from the present study. As patients of different ages may have a different prevalence of such medication use, this exclusion criterion might have caused further selection bias. Lastly, we did not have access to data regarding the types of hypertension medication used by patients, such as calcium blockers, which might have affected the findings. Further studies with other cohorts are required to validate the present findings.

## 5. Conclusions

Alcohol consumption of ≥70 g/week in male patients aged <50 years and alcohol consumption of ≥140 g/week in female patients were risk factors for EE development. Additionally, the risk of new onset of EE increased with increased drinking frequency among men and women aged <50 years. Although the influence of alcohol consumption on EE was different between men and women, age and drinking status such as quantity and frequency should be assessed when considering the prevention of EE.

## Figures and Tables

**Figure 1 nutrients-14-04760-f001:**
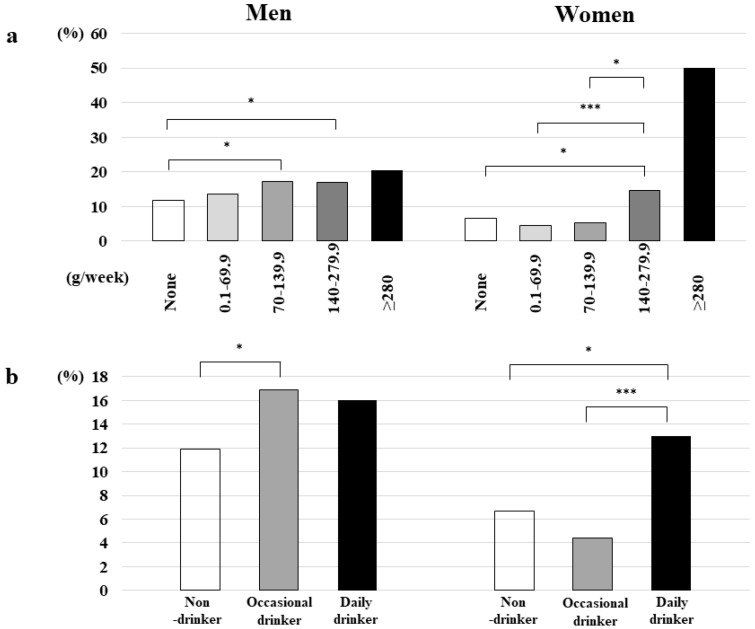
(**a**) Prevalence of new EE onset according to drinking quantity categories and sex. * *p* < 0.05, *** *p* < 0.005. EE, erosive esophagitis. (**b**) Prevalence of new EE onset according to drinking frequency categories and sex. * *p* < 0.05, *** *p* < 0.005. EE, erosive esophagitis.

**Table 1 nutrients-14-04760-t001:** Baseline characteristics according to sex.

		Total Participants	Men	Women	*p*-Value
Number	n (%)	2582	1405 (54.4)	1177 (45.6)	
Age	(years)	52.6 ± 9.0 (25–80)	52.7 ± 9.1 (25–80)	52.4 ± 8.9 (26–77)	0.317
BMI	(kg/m^2^)	23.2 ± 3.7 (14–48)	24.0 ± 3.4 (15–42)	22.2 ± 3.7 (14–48)	<0.001
WC	(cm)	82.4 ± 9.8 (56–131)	84.6 ± 9.1 (58–126)	79.9 ± 9.9 (56–131)	<0.001
Current smokers	n (%)	412 (16.0)	366 (26.0)	46 (3.9)	<0.001
Drinkers	n (%)	1504 (58.2)	1044 (74.3)	460 (39.1)	<0.001
Alcohol quantity (g/week)					
None	n (%)	1078 (41.8)	361 (25.7)	717 (60.9)	<0.001
0.1–69.9		545 (21.1)	277 (19.7)	268 (22.8)	
70–139.9		355 (13.7)	244 (17.4)	111 (9.4)	
140–279.9		571 (22.1)	493 (35.1)	78 (6.6)	
≥280		33 (1.3)	30 (2.1)	3 (0.3)	
Drinking frequency					
Non-drinker	n (%)	1078 (41.8)	361 (25.7)	717 (60.9)	<0.001
Occasional drinker		841 (32.6)	495 (35.2)	346 (29.4)	
Daily drinker		663 (25.7)	549 (39.1)	114 (9.7)	
Regular exercise	n (%)	671 (26.0)	454 (32.3)	217 (18.4)	<0.001
Eating before going to bed	n (%)	826 (32.0)	527 (37.5)	299 (25.4)	<0.001
Custom of having breakfast	n (%)	295 (11.4)	180 (12.8)	115 (9.8)	<0.05
SBP	(mmHg)	123.9 ± 16.9 (80–204)	127.3 ± 16.0 (87–204)	119.8 ± 17.1 (80–194)	<0.001
DBP	(mmHg)	78.1 ± 12.0 (43–128)	81.6 ± 11.5 (46–128)	73.9 ± 11.0 (43–116)	<0.001
Hypertension	n (%)	1208 (46.8)	796 (56.7)	412 (35.0)	<0.001
T-CHO	(mg/dL)	213.0 ± 35.1 (112–375)	208.5 ± 33.8 (112–375)	218.5 ± 35.8 (121–339)	<0.001
TG	(mg/dL)	110.5 ± 85.2 (26–1338)	130.8 ± 103.5 (28–1338)	86.2 ± 45.4 (26–423)	<0.001
HDL-C	(mg/dL)	67.1 ± 18.0 (26–145)	60.5 ± 15.8 (26–137)	74.9 ± 17.4 (28–145)	<0.001
LDL-C	(mg/dL)	129.3 ± 31.8 (29–298)	128.3 ± 31.6 (29–298)	130.5 ± 32.1 (48–247)	0.151
Dyslipidemia	n (%)	718 (27.8)	500 (35.6)	218 (18.5)	<0.001
FPG	(mg/dL)	98.0 ± 16.7 (63–305)	101.1 ± 17.7 (63–267)	94.4 ± 14.5 (67–305)	<0.001
HbA1c	(%)	5.6 ± 0.55 (4.7–13.8)	5.7 ± 0.57 (4.7–11.2)	5.6 ± 0.53 (4.7–13.8)	0.256
IGT	n (%)	1098 (42.5)	701 (49.9)	397 (33.7)	<0.001
UA	(mg/dL)	5.3 ± 1.4 (2–11)	6.0 ± 1.2 (2–10)	4.5 ± 1.0 (2–11)	<0.001
ALT	(IU/L)	22.8 ± 14.7 (5–178)	27.0 ± 16.6 (6–178)	17.7 ± 9.9 (5–125)	<0.001
AST	(IU/L)	23.9 ± 8.6 (8–126)	25.7 ± 9.5 (10–126)	21.8 ± 6.9 (8–101)	<0.001
GGT	(IU/L)	37.7 ± 40.7 (7–581)	50.3 ± 48.2 (9–581)	22.6 ± 21.1 (7–414)	<0.001
HS-CRP	(mg/L)	1.05 ± 2.97 (0.1–42.5)	1.26 ± 3.61 (0.1–42.5)	0.81 ± 1.95 (0.1–29.5)	<0.001
*H. pylori* positivity	n (%)	561 (21.7)	335 (23.8)	226 (19.2)	<0.005

Data are presented as means ± standard deviations (ranges) for continuous variables and counts (%) for categorical variables. *p*-values are based on the χ^2^ test or Mann–Whitney U test. Significance is indicated at the 5% level. ALT, alanine aminotransferase; AST, aspartate aminotransferase; BMI, body mass index; DBP, diastolic blood pressure; FPG, fasting plasma glucose; GGT, gamma-glutamyl transpeptidase; HbA1c, hemoglobin A1c; HDL-C, high-density lipoprotein cholesterol; *H. pylori*, *Helicobacter pylori*; HS-CRP, high-sensitivity C-reactive protein; IGT, impaired glucose tolerance; LDL-C, low-density lipoprotein cholesterol; SBP, systolic blood pressure; T-CHO, total cholesterol; TG, triglyceride; UA, uric acid; WC, waist circumference.

**Table 2 nutrients-14-04760-t002:** Change in alcohol quantity between baseline and final measurements (n = 2582) according to sex.

**Men (n = 1405)**							
Baseline stage			Most recent stage				
		Number	Non- drinkers	Drinkers			
	Alcohol quantity (g/week)		None	0.1–69.9	70–139.9	140–279.9	≥280
Non-drinkers	None	361 (25.7%)	335 (92.8%)	21 (5.8%)	2 (0.6%)	3 (0.8%)	0 (0%)
Drinkers	0.1–69.9	277 (19.7%)	71 (25.6)	133 (48.0%)	63 (22.7%)	10 (3.6%)	0 (0%)
	70–139.9	244 (17.4%)	15 (6.1%)	43 (17.6%)	127 (52.0%)	59 (24.2%)	0 (0%)
	140–279.9	493 (35.1%)	13 (2.6%)	14 (2.8%)	79 (16.0%)	366 (74.2%)	21 (4.3%)
	≥280	30 (2.1%)	2 (6.7%)	1 (3.3%)	2 (6.7%)	12 (40.0%)	13 (43.3%)
**Women (n = 1177)**							
Baseline stage			Most recent stage				
		Number	Non- drinkers	Drinkers			
	Alcohol quantity (g/week)		None	0.1–69.9	70–139.9	140–279.9	≥280
Non-drinkers	None	717 (60.9%)	665 (92.7%)	39 (5.4%)	11 (1.5%)	2 (0.3%)	0 (0%)
Drinkers	0.1–69.9	268 (22.8%)	69 (25.7)	154 (57.5%)	41 (15.3%)	4 (1.5%)	0 (0%)
	70–139.9	111 (9.4%)	11 (9.9%)	37 (33.3%)	50 (45.0%)	13 (11.7%)	0 (0%)
	140–279.9	78 (6.6%)	2 (2.6%)	7 (9.0%)	14 (17.9%)	54 (69.2%)	1 (1.3%)
	≥280	3 (0.3%)	0 (0%)	0 (0%)	0 (0%)	2 (66.7%)	1 (33.3%)

**Table 3 nutrients-14-04760-t003:** Change in alcohol drinking frequency between baseline and final measurements (n = 2582) according to sex.

**Men (n = 1405)**					
Baseline stage			Most recent stage		
		Number	Non-drinkers	Drinkers	
	Drinking frequency		Non-drinking	Occasional drinking	Daily drinking
Non-drinkers	Non-drinking	361 (25.7%)	335 (92.8%)	22 (6.1%)	4 (1.1%)
Drinkers	Occasional drinking	495 (35.2%)	83 (16.8)	349 (70.5%)	63 (12.7%)
	Daily drinking	549 (39.1%)	18 (3.3%)	66 (12.0%)	465 (84.7%)
**Women (n = 1177)**					
Baseline stage			Most recent stage		
		Number	Non-drinkers	Drinkers	
	Drinking frequency		Non-drinking	Occasional drinking	Daily drinking
Non-drinkers	Non-drinking	717 (60.9%)	665 (92.7%)	48 (6.7%)	4 (0.6%)
Drinkers	Occasional drinking	346 (29.4%)	75 (21.7)	249 (72.0%)	22 (6.4%)
	Daily drinking	114 (9.7%)	7 (6.1%)	18 (15.8%)	89 (78.1%)

**Table 4 nutrients-14-04760-t004:** Presence or absence of EE at the final measurement in patients without EE at baseline.

**Men (n = 1405)**						
Baseline stage		Most recent stage				
		EE (−)	EE (+)			
			Grade A	Grade B	Grade C	Grade D
EE (−)	1405 (100%)	1194 (85.0%)	182 (13.0%)	24 (1.7%)	4 (0.3%)	1 (0.1%)
**Women (n = 1177)**						
Baseline stage		Most recent stage				
		EE (−)	EE (+)			
			Grade A	Grade B	Grade C	Grade D
EE (−)	1177 (100%)	1098 (93.3%)	72 (6.1%)	7 (0.6%)	0 (0%)	0 (0%)

EE, erosive esophagitis.

**Table 5 nutrients-14-04760-t005:** Relationship between EE development and alcohol quantity according to sex.

	Alcohol Quantity (g/Week)	OR (95% CI)	OR ^a^ (95% CI)	OR ^b^ (95% CI)	OR ^c^ (95% CI)	OR ^d^ (95% CI)
Men (n = 1405)						
Non-drinkers	None	1	1	1	1	1
Drinkers		1.246 (0.908–1.709)	1.277 (0.930–1.753)	1.208 (0.879–1.659)	1.234 (0.899–1.693)	1.252 (0.907–1.726)
	0.1–69.9	0.938 (0.595–1.476)	0.938 (0.595–1.476)	0.929 (0.589–1.466)	0.952 (0.604–1.502)	0.971 (0.609–1.547)
	70–139.9	1.420 (0.947–2.128)	1.420 (0.947–2.128)	1.384 (0.921–2.078)	1.428 (0.951–2.144)	1.443 (0.954–2.182)
	140–279.9	1.275 (0.894–1.817)	1.275 (0.894–1.817)	1.228 (0.862–1.749)	1.239 (0.867–1.769)	1.255 (0.875–1.799)
	≥280	1.680 (0.720–3.923)	1.680 (0.720–3.923)	1.591 (0.681–3.717)	1.677 (0.715–3.934)	2.014 (0.848–4.787)
<50 years (n = 468)						
Non-drinkers	None	1	1	1	1	1
Drinkers		2.178 (1.125–4.217)	2.205 (1.126–4.316)	2.128 (1.094–4.140)	2.171 (1.124–4.190)	2.312 (1.171–4.564)
	0.1–69.9	1.073 (0.442–2.608)	1.131 (0.458–2.791)	1.052 (0.427–2.590)	1.085 (0.448–2.631)	1.154 (0.461–2.885)
	70–139.9	2.506 (1.175–5.346)	2.665 (1.232–5.764)	2.431 (1.135–5.209)	2.581 (1.205–5.527)	2.878 (1.302–6.361)
	140–279.9	2.609 (1.284–5.301)	2.525 (1.222–5.221)	2.559 (1.260–5.200)	2.559 (1.255–5.218)	2.666 (1.283–5.542)
	≥280	3.613 (1.004–13.00)	3.621 (0.990–13.24)	3.869 (1.056–14.17)	3.624 (1.015–12.94)	4.549 (1.228–16.85)
≥50 years (n = 937)						
Non-drinkers	None	1	1	1	1	1
Drinkers		1.003 (0.696–1.446)	1.047 (0.723–1.517)	0.979 (0.679–1.413)	0.987 (0.685–1.423)	1.009 (0.694–1.466)
	0.1–69.9	0.959 (0.565–1.628)	0.989 (0.578–1.693)	0.966 (0.569–1.639)	0.967 (0.566–1.650)	0.989 (0.572–1.709)
	70–139.9	1.122 (0.683–1.841)	1.151 (0.699–1.898)	1.112 (0.676–1.827)	1.122 (0.682–1.847)	1.134 (0.682–1.883)
	140–279.9	0.961 (0.633–1.457)	1.010 (0.662–1.541)	0.923 (0.609–1.399)	0.928 (0.614–1.404)	0.945 (0.621–1.438)
	≥280	1.059 (0.314–3.576)	1.220 (0.350–4.248)	0.934 (0.274–3.182)	1.025 (0.304–3.458)	1.227 (0.354–4.255)
Women (n = 1177)						
Non-drinkers	None	1	1	1	1	1
Drinkers		0.956 (0.604–1.512)	1.059 (0.666–1.683)	0.983 (0.612–1.580)	1.003 (0.634–1.586)	1.078 (0.666–1.747)
	0.1–69.9	0.630 (0.327–1.212)	0.686 (0.355–1.327)	0.650 (0.337–1.257)	0.648 (0.337–1.249)	0.692 (0.353–1.356)
	70–139.9	0.768 (0.327–1.800)	0.847 (0.360–1.992)	0.806 (0.338–1.923)	0.818 (0.346–1.931)	0.911 (0.375–2.210)
	140–279.9	2.195 (1.141–4.221)	2.755 (1.393–5.449)	2.399 (1.214–4.741)	2.427 (1.261–4.670)	3.040 (1.504–6.146)
	≥280	7.152 (1.265–40.45)	11.18 (1.830–68.29)	10.18 (1.252–82.73)	7.531 (1.367–41.50)	13.26 (1.735–101.4)
<50 years (n = 413)						
Non-drinkers	None	1	1	1	1	1
Drinkers		1.626 (0.716–3.694)	1.638 (0.704–3.811)	1.719 (0.735–4.018)	1.687 (0.724–3.927)	1.764 (0.736–4.231)
	0.1–69.9	1.024 (0.326–3.212)	1.017 (0.319–3.237)	1.056 (0.336–3.323)	1.046 (0.324–3.382)	1.072 (0.328–3.505)
	70–139.9	1.604 (0.437–5.884)	1.617 (0.450–5.808)	1.865 (0.480–7.242)	1.584 (0.425–5.906)	1.873 (0.481–7.299)
	140–279.9	2.553 (0.805–8.093)	2.708 (0.789–9.291)	3.014 (0.876–10.37)	2.810 (0.904–8.736)	3.074 (0.940–10.05)
	≥280	22.03 (4.919–98.62)	24.21 (5.891–99.46)	33.96 (3.184–362.1)	19.90 (6.319–62.64)	30.87 (3.693–258.1)
≥50 years (n = 764)						
Non-drinkers	None	1	1	1	1	1
Drinkers		0.759 (0.429–1.344)	0.885 (0.497–1.576)	0.747 (0.414–1.347)	0.788 (0.447–1.389)	0.853 (0.461–1.581)
	0.1–69.9	0.516 (0.230–1.162)	0.594 (0.261–1.350)	0.526 (0.232–1.193)	0.530 (0.236–1.192)	0.589 (0.253–1.370)
	70–139.9	0.519 (0.161–1.675)	0.606 (0.185–1.986)	0.528 (0.159–1.751)	0.555 (0.171–1.799)	0.619 (0.182–2.110)
	140–279.9	2.285 (1.026–5.088)	2.928 (1.281–6.690)	2.209 (0.960–5.079)	2.425 (1.075–5.470)	3.149 (1.230–8.064)
	≥280	(−)	(−)	(−)	(−)	(−)

^a^ OR adjusted for age, BMI, and WC. ^b^ OR adjusted for lifestyle habits. ^c^ OR adjusted for hypertension, dyslipidemia, and IGT. ^d^ OR adjusted for age, BMI, WC, lifestyle habits, hypertension, dyslipidemia, IGT, and *H. pylori*. (−) means the absence of participants. BMI, body mass index; CI, confidence interval; EE, erosive esophagitis; *H. pylori*, *Helicobacter pylori*; IGT, impaired glucose tolerance; OR, odds ratio; WC, waist circumference.

**Table 6 nutrients-14-04760-t006:** Relationship between EE development and drinking frequency according to sex.

	Drinking Frequency	OR (95% CI)	OR ^a^ (95% CI)	OR ^b^ (95% CI)	OR ^c^ (95% CI)	OR ^d^ (95% CI)
Men (n = 1405)						
Non-drinkers	None	1	1	1	1	1
Drinkers	Occasional	1.259 (0.880–1.802)	1.298 (0.904–1.863)	1.236 (0.863–1.771)	1.263 (0.882–1.809)	1.294 (0.896–1.867)
	Daily	1.234 (0.871–1.748)	1.261 (0.888–1.789)	1.184 (0.836–1.676)	1.209 (0.854–1.713)	1.215 (0.853–1.731)
<50 years (n = 468)						
Non-drinkers	None	1	1	1	1	1
Drinkers	Occasional	1.879 (0.926–3.815)	1.929 (0.938–3.966)	1.854 (0.909–3.782)	1.897 (0.938–3.836)	2.051 (0.986–4.268)
	Daily	2.588 (1.278–5.242)	2.597 (1.259–5.353)	2.515 (1.237–5.112)	2.536 (1.252–5.136)	2.692 (1.298–5.586)
≥50 years (n = 937)						
Non-drinkers	None	1	1	1	1	1
Drinkers	Occasional	1.116 (0.728–1.710)	1.151 (0.745–1.778)	1.113 (0.726–1.705)	1.105 (0.719–1.699)	1.146 (0.738–1.779)
	Daily	0.930 (0.619–1.395)	0.978 (0.649–1.474)	0.892 (0.595–1.339)	0.910 (0.608–1.361)	0.917 (0.609–1.381)
Women (n = 1177)						
Non-drinkers	None	1	1	1	1	1
Drinkers	Occasional	0.610 (0.336–1.106)	0.668 (0.368–1.213)	0.633 (0.345–1.160)	0.639 (0.352–1.158)	0.690 (0.374–1.273)
	Daily	1.996 (1.121–3.554)	2.329 (1.281–4.234)	2.109 (1.159–3.838)	2.121 (1.193–3.768)	2.444 (1.320–4.524)
<50 years (n = 413)						
Non-drinkers	None	1	1	1	1	1
Drinkers	Occasional	1.069 (0.395–2.895)	1.050 (0.387–2.852)	1.127 (0.412–3.079)	1.094 (0.397–3.020)	1.147 (0.409–3.220)
	Daily	3.378 (1.234–9.247)	3.721 (1.275–10.86)	3.892 (1.288–11.76)	3.620 (1.343–9.755)	4.030 (1.404–11.57)
≥50 years (n = 764)						
Non-drinkers	None	1	1	1	1	1
Drinkers	Occasional	0.472 (0.219–1.016)	0.546 (0.251–1.185)	0.484 (0.222–1.056)	0.490 (0.228–1.056)	0.553 (0.249–1.230)
	Daily	1.603 (0.783–3.282)	1.957 (0.936–4.091)	1.522 (0.724–3.197)	1.676 (0.819–3.431)	1.860 (0.833–4.154)

^a^ OR adjusted for age, BMI, and WC. ^b^ OR adjusted for lifestyle habits. ^c^ OR adjusted for hypertension, dyslipidemia, and IGT. ^d^ OR adjusted for age, BMI, WC, lifestyle habits, hypertension, dyslipidemia, IGT, and *H. pylori*. BMI, body mass index; CI, confidence interval; EE, erosive esophagitis; *H. pylori*, *Helicobacter pylori*; IGT, impaired glucose tolerance; OR, odds ratio; WC, waist circumference.

## Data Availability

The data that support the findings of this study are available from the corresponding author upon reasonable request.

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
