# Peer review of "Influence of Alcohol Consumption on the Development of Erosive Esophagitis in Both Sexes: A Longitudinal Study"

_nutrients, 2022, doi:10.3390/nu14224760_

Round 1

Reviewer 1 Report

Sogabe et al. studied influence of alcohol consumption on the development of erosive esophagitis.

My remarks about the paper:

-the data were probably collected about the type of alcohol used (as mentioned: beer, Japanese sake, shochu,  wine,  whisky) but then authors showed only the influence of amount of alcohol on consecutive parameters and not type of it. If kind of alcohol could be implemented in the paper, it would valuable.

-authors  hypothesized that the risk of EE development would increase with an increase in alcohol consumption… but than in conclusions the resume:…among men and women aged <50 years, alcohol consumption did not have a protective effect against new onset of EE. No possible protective effect was hypothesized before- so I would just remove the first sentence from conclusions.

-the second paragraph of discussion (about Hp infection):  I would move down in this section- this is just co-finding in this study and interrupts the discussion.

-some minor English corrections to be performed

To sum up I recommend the revisions suggested above.

Author Response

Response to reviewer’s comments

(A point-by-point response)

Reviewer 1

Sogabe et al. studied influence of alcohol consumption on the development of erosive esophagitis.

My remarks about the paper:

-the data were probably collected about the type of alcohol used (as mentioned: beer, Japanese sake, shochu,  wine,  whisky) but then authors showed only the influence of amount of alcohol on consecutive parameters and not type of it. If kind of alcohol could be implemented in the paper, it would valuable.

Response: Thank you for the suggestion.

Participants in the present study were all Japanese, and the numbers of participants who drunk wine and whisky was very small. Therefore, analysis according to the type of alcohol that was consumed would be difficult. We apologize that we cannot implement your suggestion.

-authors  hypothesized that the risk of EE development would increase with an increase in alcohol consumption… but than in conclusions the resume:…among men and women aged <50 years, alcohol consumption did not have a protective effect against new onset of EE. No possible protective effect was hypothesized before- so I would just remove the first sentence from conclusions.

Response: Thank you for the constructive comment.

Instead of deleting the first sentence, we added the following sentence to the conclusions section (Lines 312-313): “Alcohol consumption of ≥70 g/week in male patients aged <50 years and alcohol consumption of ≥140 g/week in female patients were risk factors for EE development.”

-the second paragraph of discussion (about Hp infection):  I would move down in this section- this is just co-finding in this study and interrupts the discussion.

Response: Thank you for the suggestion.

Indeed, the main contents of the present study were the influence of alcohol consumption on the development of erosive esophagitis. However, H. pylori was a significant factor for the development of erosive esophagitis. Hence, we reported this result in the present study.

-some minor English corrections to be performed

Response: Thank you for pointing this out.

We asked an English proofreading company (already listed in the Acknowledgments section as Editage) to proofread the revised manuscript.

To sum up I recommend the revisions suggested above.

Submission Date

22 September 2022

Date of this review

25 Oct 2022 11:57:35

Reviewer 2 Report

The authors have investigated the association between alcohol consumption and development of erosive esophagitis using longitudinal data from a population that underwent frequent comprehensive health examinations. The population was large and the statistical methods well described and appropriate. The main concerns are related to how the selection of the 2582 participants from the 13448 individuals to begin with influence the interpretation of the results. Consequently there is a question of validity of the results as well as if there the described associations do have a clinical relevance.  The recommendation of no alcohol consumption for individuals aged <50 years to avoid erosive esophagitis development is formally correct, but the underlying data too weak for such a specific recommendation.

Major remarks:

Lines 48-50: the formulation of the hypothesis is a bit unclear. What did you expect in younger patients compared to older patients? And why? Please refer to studies that may have led to the hypothesis you formed.

Line 61-63: how did you handle patients in the study who started acid inhibiting medication between upper endoscopies, either against symptoms of GERD or against other disease? How many patients were these? Were they excluded?

It seems remarkable that alcohol should be a risk factor for EE only in individuals < 50 years. One explanation may be patient selection caused by the exclusion criteria concerning PPI/H2-blocker use. If many PPI users > 50 years were excluded before the study, this could explain the observed phenomenon. Please mention in the discussion.

Please state the number of participants excluded due to the various causes listed in lines 58-65.

The most common cause of erosive esophagitis is gastroesophageal junction incompetence, by either insufficiency of the lower esophageal sphincter or a hiatus hernia. Does alcohol consumption reduce the pressure of the lower esophageal sphincter? 

Minor:

Regular exercise was defined as >30 minutes at least once weekly, this is a very modest level of activity with little health benefit. Do the authors have any references or explanation that could be added to the Methods section?

Table 1. Variables that do not have normal distribution should be presented as median followed by range or interquartile range, rather than mean (SD).

Author Response

Response to reviewer’s comments

(A point-by-point response)

Reviewer 2

Comments and Suggestions for Authors

The authors have investigated the association between alcohol consumption and development of erosive esophagitis using longitudinal data from a population that underwent frequent comprehensive health examinations. The population was large and the statistical methods well described and appropriate. The main concerns are related to how the selection of the 2582 participants from the 13448 individuals to begin with influence the interpretation of the results. Consequently there is a question of validity of the results as well as if there the described associations do have a clinical relevance. The recommendation of no alcohol consumption for individuals aged <50 years to avoid erosive esophagitis development is formally correct, but the underlying data too weak for such a specific recommendation.

Major remarks:

Lines 48-50: the formulation of the hypothesis is a bit unclear. What did you expect in younger patients compared to older patients? And why? Please refer to studies that may have led to the hypothesis you formed. 

Response: Thank you for your comment.

We described the detailed reasons for our hypothesis with references in the discussion section as we wanted to avoid a long explanation in the introduction section. Please see Lines 280-286.

Line 61-63: how did you handle patients in the study who started acid inhibiting medication between upper endoscopies, either against symptoms of GERD or against other disease? How many patients were these? Were they excluded? 

Response: Thank you for the questions.

We excluded patients in the study who started acid-inhibiting medication between the initial stage and the most recent stage. This is described in the Materials and Methods section (Design and study population subsection) (Lines 59-64: “At baseline and in the final cohort...”).

We showed the number of excluded patients in Figure S1 (the flow diagram).

First, we excluded patients who underwent health check-up only once during the study period.

Seconds, we excluded patients who had upper intestinal diseases such as GERD, peptic ulcer, upper intestinal cancer, and so on.

Thirds, we excluded patients who took acid-inhibiting medication, such as H2-receptor antagonists and proton pump inhibitors because they took antiplatelet drug or steroid drug or NSAIDs due to ischemic heart disease, cerebral vascular disease, collagen disease, orthopedic disease and so on. The number was 51 patients.

Fourth, we excluded patients who were diagnosed with upper gastrointestinal cancer at the time of EGD.

We investigated diseases such as GERD, peptic ulcer, upper intestinal cancer, and so on using a self-report questionnaire. We investigated the use of acid-inhibiting medication, such as H2-receptor antagonists and proton pump inhibitors, and during follow-up using a self-report questionnaire.

According to your comments, we changed the sentence (Lines 73-75) from “The following information was obtained using a self-report questionnaire: smoking status, alcohol intake, exercise, meals, and drug history.” to “The following information was obtained using a self-report questionnaire: smoking status, alcohol intake, exercise, meals, drug history (including acid-inhibiting medication), and history of the present illness.

It seems remarkable that alcohol should be a risk factor for EE only in individuals < 50 years. One explanation may be patient selection caused by the exclusion criteria concerning PPI/H2-blocker use. If many PPI users > 50 years were excluded before the study, this could explain the observed phenomenon. Please mention in the discussion. 

Response: We agree with your comments.

We added a sentence about the influence of excluding patients who used PPIs before the present study in the revised discussion section. Specifically, we changed a sentence (Lines 305-310) from “Lastly, although we collected data regarding the history of medications such as proton pump inhibitors and H2-receptor antagonists, we could not investigate the content of hypertension medication, such as calcium blockers, that may affect the findings.” to “Fourth, we excluded patients taking acid-inhibiting medications from the present study. As patients of different ages may have a different prevalence of such medication use, this exclusion criterion might have caused further selection bias. Lastly, we did not have access to data regarding the types of hypertension medication used by patients, such as calcium blockers, which might have affected the findings” in the explanation of limitations according to your comments.

Please state the number of participants excluded due to the various causes listed in lines 58-65.

Response: Thank you for the request.

The numbers of excluded participants are provided in Figure S1 (the flow diagram).

We excluded participants who had incomplete information; those who underwent previous digestive tract surgery, including EMR and ESD for gastric cancer, gastric adenoma, gastric SMT, early SCC, duodenal polyps, and gastroduodenal bleeding ulcer; those who visited the hospital for treatment or were followed up for upper intestinal diseases, such as GERD, peptic ulcer, past peptic bleeding ulcer; those with upper intestinal cancer, including those on chemotherapy; those who were taking medications such as H2-receptor antagonists, proton pump inhibitors, and medications to accelerate gastrointestinal peristaltic motion; those who were taking acid-inhibiting medications because of taking antiplatelet agents; and those who were diagnosed with upper gastrointestinal cancers such as gastric cancer and suspected SCC at the time of esophagogastroduodenoscopy. As there were so many types of causes for exclusion, only the total number of excluded participants were listed in the flow diagram.

The most common cause of erosive esophagitis is gastroesophageal junction incompetence, by either insufficiency of the lower esophageal sphincter or a hiatus hernia. Does alcohol consumption reduce the pressure of the lower esophageal sphincter? 

Response: This is an insightful comment.

We had the same line of thought on this issue. We believe that the extension of the acid exposure time owing to alcoholic carbonated beverages may induce erosive esophagitis. However, we cannot demonstrate this expectation in the present study. Therefore, we did not discuss it in our manuscript.

Minor:

Regular exercise was defined as >30 minutes at least once weekly, this is a very modest level of activity with little health benefit. Do the authors have any references or explanation that could be added to the Methods section? 

Response: Thank you for the comment.

We utilized a routine questionnaire used in our facility. We apologize for not obtaining information regarding higher levels of activity.

Table 1. Variables that do not have normal distribution should be presented as median followed by range or interquartile range, rather than mean (SD). 

Response: Thank you for pointing this out.

Accordingly, we added the ranges to Table 1 in the revised manuscript.

Submission Date

22 September 2022

Date of this review

28 Oct 2022 13:03:47

Round 2

Reviewer 2 Report

I thank the authors for the revision. The study is comprehensive, the limitations have been discussed appropriately and further improvements do not seem possible.